# Downside: The Perpetrator of Violence in the Representations of Social and Health Professionals

**DOI:** 10.3390/ijerph17197061

**Published:** 2020-09-27

**Authors:** Fortuna Procentese, Roberto Fasanelli, Stefania Carnevale, Ciro Esposito, Noemi Pisapia, Caterina Arcidiacono, Immacolata Di Napoli

**Affiliations:** 1Department of Humanities, University of Naples “Federico II”, 80133 Napoli, Italy; fortuna.procentese@unina.it (F.P.); ciro.esposito5@unina.it (C.E.); pisapia.noemi@gmail.com (N.P.); caterina.arcidiacono@unina.it (C.A.); immacolata.dinapoli@unina.it (I.D.N.); 2Department of Sociology, University of Naples “Federico II”, 80133 Napoli, Italy; roberto.fasanelli@unina.it

**Keywords:** gender-based violence, perpetrator repetitiveness, health and welfare services, representation, taking charge of violence

## Abstract

Gender-based violence is a widespread phenomenon and pandemic that affects women’s lives. Many interventions have been activated for perpetrators, but the dropout rate is still high. In order to draw up guidelines for responsibly and sustainably dealing with the phenomenon, this study is aimed at investigating the professionals’ perception of the perpetrator as a useful element in designing innovative intervention policies. Open interviews were carried out with welfare and health professionals and the Grounded Theory Methodology was used to analyze the collected data. These results detect attitudes of social health personnel and their feelings of impotence towards gender-based perpetrators because of the emergence of an inevitable repetitiveness of the violent behavior, as well as the “normality of violence” in a patriarchal culture and its “transversality”. This reflective knowledge allows for the opportunity to develop best transformative attitudes toward the phenomenon. According to the results, it is urgent to establish an active and convinced alliance with the healthy part of the man, through specific prevention paths, in order to activate an authentic motivation for change and its sustainability.

## 1. Introduction

Gender-based violence is a widespread phenomenon, and the WHO [1] stated that it is a pandemic that affects women’s lives. International organizations have focused on this phenomenon, describing its features and the need for preventative measures, emergency actions and recovery interventions that countries have to undertake, as stated by the Istanbul Conference and its applications [2,3,4,5]. There is an increased knowledge about women in their role as victims, and measures to contrast gender-based violence against women are at play. However, the interest for the perpetrator is a most recent issue and specific program directed to them have only recently been organized in different countries [6]. A recent evaluation of European programs for perpetrators has identified the guidelines to be adopted in these programs, suggesting, in particular, the need for an ecological approach in taking charge of the perpetrator [7].

In Italy, among the 133 women killed in 2018, 81.2 percent were killed by a very close person and the term attributed to this crime is femicide [8], and Istat’s investigation [9] *Stereotypes about Gender Roles and the Social Image of Sexual Violence* presents data on the number of people reported, the victims of persecutory acts, abuse in the family, beatings, and sexual violence, but in the social reality of everyday life the phenomenon is actually still invisible [10]. In fact, most of the violence against women occurs in the home [11], causing physical, psychological and economic damage [12,13,14,15,16,17].

There is extensive literature on IPV (Intimate Partner Violence) victims [2,3,4,5,18], while the studies on the authors of violence and their treatment are still to be investigated. Moreover, recent studies have focused on the symbolic and valuable universe of professionals involved in the treatment of violence, as a fundamental dimension in dealing with the phenomenon [10,19,20]. Indeed, in the Italian context [21], a high rate of dropout of treatments directed towards men, the frequent breakdown of dedicated services, the scarcity of dialogue and the inadequate training of health and welfare professionals further perpetrate the pain and the suffering of direct and collateral victims and actors [10]. An in-depth study of these theoretical dimensions is necessary in order to guarantee an integrated and efficient system to deal with gender-based violence.

### 1.1. Literature Review

#### 1.1.1. The Roots of Violence Inflicted by Men

Bourdieu [22] calls “symbolic power” the *unconscious social reproduction of male culture*, which is accepted by both the dominated and the dominant as the natural state of things [23,24]. Specifically, the gender perspective focuses on these dynamics of power and sees the man as the holder of power within the relationship and therefore his violent behavior would be aimed at the control and possession of the partner or ex-partner [21,22]. Conversely, on an individual level, perpetrators are often represented by frailty, pathological narcissism and difficulties in communication and reasoning [25,26], lack of emotional self-regulation [27,28] and interpersonal skills [29,30]. Richard Mizen [31] states that “violence does not represent an absence of the mind, but rather an ablation of the mind in which the individual struggles with affective experiences that he is unable to manage, as well as not being able to recognize their important mental qualities through defence psychology” (p. 75). Therefore, violent behavior may be characterized by the subjective psychological and affective substrates of violence that are brought out. A sense of self-annihilation and helplessness that lead to intolerable and unbearable fears, such as the experience of pervasive panic, is the backdrop for violent acts. In these cases, individuals deprived of their primary defenses became unable to reflect (mentalize) on their thoughts. For men in these circumstances, it is absolutely necessary to be able to control the perceived internal threats of annihilation closely related to homicidal/suicidal fantasies and activate a process of self-regulation of emotions. Therefore, in a biopsychosocial analytic approach, much of the violent behavior is inscribed in early experiences of affects and emotions [32].

Indeed, violence is an expression of the failure of the psychic integration process, which concerns the integration of emotional experiences. According to the psychoanalytic model, this failure leads men to project the divided and hostile parts of themselves (parts on which they can neither think nor elaborate) on their partners through a process of projective identification. This mechanism is used by violent adults as a psychological defense, in absence of an ability to reflect on one’s emotions. From this point of view, other people become a repository for these parts of violent adults, so that any change, departure or non-response threatens their sense of self and potentially causes self-disintegration. This mechanism creates a hostile context around men, but, at the same time, inevitably keeps men tied to their companions [33].

On an individual level, the perpetrators minimize the severity of the assault and deny the violence [34]. This silence is experienced by women as a “burden”. The silence of men denies value and identity. It is left unsaid but it says: “you are not worthy and you are nothing” and therefore “I act as if you didn’t exist”. It expresses indifference, lack of recognition, exclusion of confrontation. Silence is a form of obstructionism that undermines intimacy, trust and happiness of a relationship, leads the victim to feel wrong [35]. One of the excuses used by violent men is that they were defending themselves; moreover, men blame their partner for being jealous, unstable or in control of anger, while presenting a positive image of themselves because of their willingness to engage in social life [36].

Furthermore, the attitude attributed to men is often a devaluation of one’s own behavior, blaming the victim and denigrating the bond with her. [26,37,38]. For example, it is difficult for some to admit that they perpetrated violence as self-defense, because this admission implies vulnerability [39].

A questionable approach proposed by Johnson [40] theorized the existence of two different forms of violence in couples: (1) patriarchal terrorism, which in a feminist perspective, not only descends from cultural stereotypical burden but it is compound by different forms of violence, their escalation and interactions and (2) couple violence, which arises just from family dynamics. In this model, patriarchal terrorism is a product of social traditions whereby men feel authorized to control women through the use of violence, subordination and isolation; in our vision, however, conflicts that arise in the couple between partners are also frequently inscribed within a patriarchal frame, even when they rarely lead to serious forms of violence. Although this approach induces us to take into account the interaction and escalation of forms of violence more than its incidence and prevalence it is a way to detect the more serious cases and their danger to women’s lives, and at the same time it may be the background to create intervention strategies directed to severe violence perpetrators detecting precociously the dangers of their behavior for women and children.

Hearn [41], conversely offers a more inclusive approach that defines the causes of men’s violence against women considering three levels: The first is the individual level, the second focuses on the family and other relationships, such as socialization and learning within the family, and the last level focuses on the wider socio-cultural aspects related to power. In the same vein, international organizations propose an ecological model [42] and recently Di Napoli, et al. [10] according to a wider ecological approach, proposing an operational model that expresses the interaction of individual factors together with organizational and relational issues. The following Figure 1 summarizes the reciprocal interactions of individual, social and organizational issues: patriarchal cultural prescriptive and stereotyped role, individual feelings acting in women, children, and men, and the protective action of an institutional service system.

Referring to the treatment of perpetrators, there is also the need to let different institution and services interplay [43]. In the article “Domestic Violence Perpetrator Programs in Europe, Part II A Systematic Review of the State of Evidence” [44], the authors find that the assessment of the treatment of the authors of Domestic violence in Europe needs to be improved and programs should become more tailored to the characteristics of the participants. This perspective clashes with the vision that funds spent on men’s care are perceived as “diverted” from more urgent priorities, especially considering the need to protect women and take care of children who have been exposed to violence [33]. Differentiating motivations, reasons, functions, justifications and contexts is the challenge that researchers have to face in this sector. Some reasons may be easier to report than others.

#### 1.1.2. Social and Health Personnel Facing Gender Violence Perpetrators

In this general frame to detect new strategies and welfare measures, it is important to acquire knowledge about professionals dealing with gender-based violence, their feelings and their representation of the phenomenon. Arcidiacono and Di Napoli [45] analyzed perceptions of GP (General Practitioners), parish and social and health services personnel about gender-based violence; Creazzo [46] investigated the thoughts of different professionals on three European cases specifically on violence perpetrators. Professional reflexivity [47,48,49,50] is in fact a significant tool for understanding undergoing interventions and for supporting successful intervention strategies. Autiero et al. [43] specifically focused on the fact that health and social personnel viewpoints differ depending on the service with which they are involved [51]. In the Campania Region “In the case of CAV operators, violence management focuses on protecting women and their children from serious and immediate danger, “setting aside” perpetrators and any possible assistance that could be provided to them. However, in the case of Oltre la Violenza (OLV—a local health service dedicated to perpetrators of violence) staff members state that working with gender violence perpetrators is a priority that aims to “also care for” the “suffering” that these men have been subjected to, in order to reduce their violent behavior” (p. 13). Gender also leads to further differences in professional attitudes towards the perpetrators, as examined by Chiurazzi and Arcidiacono [52] and Autiero et al. [43], who specifically analyzed advantages, disadvantages and features of the different sex of professionals taking care of them.

Based on social and health professionals’ reports and on the protagonists of the violence, Swan et al. [53] showed that men (and when deemed necessary, also women) commit violence to regain or maintain control of their relationships, to defend themselves and in response to retaliation from previous abuse. The ‘benefits’ for the attacker include recovering a sense of power or control, protecting themselves from ongoing physical or emotional pain (i.e., self-defense), transmitting communication about intrapersonal (i.e., anger) or interpersonal (i.e., dissatisfaction with relationships, jealousy) processes, or retaliation for past injustices, or discomfort from a lack of control, or finally an inability to communicate.

The massive regression, the denial of addiction and the need to control the object often create forms of violence about which the subject does not grasp the extreme childish aspect: the expression of physical supremacy indeed serves to reassure him about his superiority over the object, on whom he is instead so dependent [54].

From a study conducted among professionals, Amodeo et al. [20] find that female staff frequently work with male authors of IPV using a “perceived listening” strategy, rather than empathy (p. 16); it is a type of listening that allows you to maintain emotional distance and a more neutral attitude towards men. Furthermore, when staff and authors belong to the same sex, this allows them to better reflect on the issues of masculinity and empathy. Indeed, Amodeo et al. [20] showed that male staff perceive the need to recognize ambivalent aspects of violence in themselves more often than female staff and this underlines the importance of contact with their emotions and emotional experiences. Consequently, male staff are more likely than women to recognize the need to develop self-reflective professional skills that allow them to maintain a “binocular vision” oriented towards themselves and others. Procentese et al. [19] focused their research on emotional experiences and on the difficulties for professionals dealing with gender-based perpetrators, highlighting the need for specialized training and the improvement of the whole services’ organization combatting violence against women. Service personnel involved in the prevention of gender-based violence consider working with men to be a great challenge. The perpetrators are seen as men who are unable to recognize their violent behavior and who deny it. This means that they must be encouraged to undertake a path of change and pushed to develop awareness of their ways of thinking and acting. These men need access to treatment, but in a “spintaneous” way (from the Italian word “spinto” (pushed) combined with the word “spontaneous”). This leads to the hypothesis that spaces for mutual assistance and a “male” space lead to greater efficiency, given that “dialogue with women seems to be an objective, not a starting point” (p. 133), and the male staff could be a resource for working with perpetrators.

As Arcidiacono and Di Napoli [45] proved, health and welfare professionals working in general services represent a point of reference for people, because they work in a well-known public institutional network.

However, it is a network to be cared for and whose well-being is of fundamental importance; the training and supervision of those who work in health and welfare services is a very important element in ensuring effective intervention and the well-being of its own workers [55].

Indeed, according to Taylor et al. [55], “working on a helpline resulted in challenges to their well-being, commonly affecting their personal lives, their relationships and their ability to cope at work” (p. 5).

Therefore, a debate about the needs of gender-based violence helplines to have supervision and support and specific training has been opened and studies exploring the perspective of staff working for helplines, including managers and policy makers, are increasing [10,19,20,43,55,56].

The aim of this study is to understand how the personnel of health and cultural centers specifically represent his/her work with IPV violence perpetrators, their feelings and thoughts. There is the need to deepen how their representation influence the vision of the phenomenon and also to understand if their visions of the perpetrators affect the services’ intervention strategies and enhance the cultural comprehension of the phenomenon. The research is situated in South Italy, a context where a patriarchal culture is still very deeply rooted in the whole men–women relationship [45]. In three companion papers, we, respectively, analyzed the professional view of women–men interaction in intimate partner violence [23], of the child witnessing violence [56] and their different viewpoints in women’s health shelters and in services geared to perpetrators of violence [43]. In these articles, we highlighted the presence of different points of view and representation among them and their difficulty in dealing with perpetrators of violence. Finally, this article will be aimed at examining the features of the perpetrator in the view and attitudes of professionals dealing with gender-based violence. Our final aim is to be able to propose suggestions to improve service organizations and make their actions more effective.

## 2. Materials and Methods

### 2.1. Participants

The participants were 50 individuals (45 women and five men), aged between 27 and 70 (SD = mean 45.56). All of them had work experience in the prevention and treatment of domestic violence, volunteers and professionals with years of service from 1 to 45 in the field of violence, covering different professional roles: Psychologists, Psychotherapists, Social Workers, Honorary Judge, CTU (Technical Consultant Office), Campania Region Councillor, ASL Managers NA 1 Centre, Family Mediators, Educator, Lawyers, Criminologists, Professional Nurse, Emergency Surgeon, Official Campania Region.

The participants were selected from cultural, health and social professionals who deal with the issue of domestic violence in the Naples area; their workplaces were: regional healthcare and social services (Clinical Psychology Operational Unit (Local health service—Naples 1), Maternal and Child Operational Unit (Local health service—Naples 1), Juvenile and Family Services of the City of Naples, Center for Families (Local health service—Naples 1), private psychology clinics, and anti-violence centers (CAV) organized by nonprofit associations and located at Cardarelli Hospital and in districts 24 and 26 of the City of Naples. Table 1 shows a wider description of participants.

The sampling strategy was not probabilistic but theoretical, aimed not at guaranteeing the representativeness of the participants in relation to the universe of the population, but best expressing knowledge and experience closely connected to the phenomenon under investigation. The theoretical sampling consists, in fact, in identifying participants that, for their role and experience, or even their life context and workplace, may have an awareness and implicit knowledge on the research area. In other words, theoretical sampling requires extending the sample to those individuals that are bringing added value to the research, while it is useless to continue it where the data are redundant [57].

The interviews were carried out at the interviewees’ workplace or at the University of Naples Federico II. They took place in a quiet and reserved environment and lasted between 30 min and 2 h, with an average of 50 min. Maximum care of the work team was given to meeting the needs of the interviewees regarding the times and dates of the appointments, offering, for this purpose, the use of the university headquarters for interviews. The interviewees signed an informed consent form and authorized the use of the data collected for research purposes; The Ethical Committee of Psychological Research, Department of Humanities at University of Naples “Federico II” gave its approval according to the Declaration of Helsinki (CERP 15b/2019–15/3/2019). Audio recordings and consent forms were stored in a locked archive.

An open interview was carried out according to the principles of Legewie [58] and Schütze [59]. The open interview is a research tool that offers the researcher the chance of develop themes of interest and also allowing the interviewee to propose themes and topics of interest for him/her.

It does not include predetermined questions, but a guide that outlines the topics to be addressed, an open “canvas” that directs the interviewer into the interaction. The structuring and administration of the interview presupposes knowledge of the subject matter, the general and specific purposes of the research and competence in conducting the interviews.

### 2.2. Procedures and Methods

To analyze the data, we used the Grounded Theory Methodology [60,61], applying it through the Atlas 8.0 software. Grounded Theory Methodology (GTM) is defined as a general method, as well as a set of tools and techniques for dealing with empirical research data. The GTM is concretized in the dialogue between the analytical rigor and rational need of methodical order, and fieldwork, with openness to experience, attention to symbolic devices and flexibility of research practices [60]. An adaptation of GTM was progressively proposed based on the generativity of local and contextual theories, rethought in constructivist [62] and situational [63] perspectives; in these perspectives, the key point is the way of understanding the role of the researcher, who is no longer considered separate from the observed object.

This methodology allows the data to “speak” and discover theories starting from the empirical research process, going beyond the already existing theories.

Indeed, after an initial more descriptive analysis of the textual material (theoretical coding), the analysis process becomes more interpretative, identifying thematic macro-categories that contain more relevant analytical meanings and that allow you to categorize the data in a more incisive and comprehensive way [62].

It consists of three types of coding phases that characterize that structure all the work on the textual material:

Open coding characterized by the fragmentation of data in sensitizing units, in order to carry out a sampling of first analysis units to be grouped into “codes” and “categories”. In this first step there is the maximum flexibility.

b. Axial coding that allows to define the relationships between the various categories; causal relationships, of membership, opposition, etc. in which the various units play roles and functions that may be also different.

c. Selective coding that provides a higher level of abstraction, leading to the identification of overdetermined categories (Macro Categories) that are more generic and that approach the formulation of the new theory [61].

So, the categorization process begins with an elementary codification which progressively takes on greater complexity and generalization. Finally, the last essential step of the Grounded Theory Methodology consists in identifying what is called the core category, the central category connected to all the others and which manages to summarize all of them.

Through an insight, it allows one to find in the data a new conceptualization, a further meaning not explainable through categorizations already carried out.

## 3. Results

The codes emerged from the whole coding process of the 50 interviews were grouped into several categories and macro categories. In this article, we will conceptualize only those (107) that specifically refer to perpetrators included in four/macro categories described below: Section 3.1 The representation of the perpetrator; Section 3.2 The representation of gender-based violence acted by perpetrators; Section 3.3 The “naturalness” of gender-based violence of perpetrators; Section 3.4 The possibility of change for gender-based violence perpetrators.

### 3.1. The Representation of the Perpetrator

The categories grouped within this conceptual macro-area concerning how the perpetrator is described and experienced by interviewees are:

Personality traits and justification of violence: emergent characteristics of the perpetrator.

In the personnel’s words, the perpetrators always present a full justification of their violent acts:

“… Then they say things like: ‘Yes, but she provoked me’, ‘Yes, but she told me to’, ‘Yes, I did it, but because I am jealous, because I love her.” (I_21—Gender: F, age: 60, profession: sociologist, workplace: CAV).

But at the same time, they show emotional dependence and sometimes drug addiction is associated with their dependence on their partner.

They are also highly disparaging towards their partner:

“He needs to denigrate the other to feel superior himself. So, a sort of psychological game of denigrating the other is created.” (I_10—Gender: M, age: 49, profession: officer, working context: municipality).

In the interviewees’ vision, men often present:Difficulty Expressing Themselves and Their Emotions

“Their way of expressing their aggression, their emotions, even their pain and frustrations, takes place through acting out, because they had no possibility to express their feeling differently except through actions, therefore, that transitional passage of emotional signification, of putting emotions into words is missing” (I-02—Gender: F, age: 34, profession: psychologist, workplace: CAV).

Compulsive Obsessive Traits

“Because he owns her, because he reifies her, she becomes his object. That is convenient for them: they do not participate in the relationship, but they (men) fulfil the need to be placed at the center of attention. So, in general they are very demanding people, who like to be placed at the center of attention … they are also very narcissistic, they are manipulators … and they do not easily give up this privileged status.” (I_06—Gender: F, age: 55, profession: doctor, work context: hospital, emergency unit).

Manipulative Behavior

“There is a real mental manipulation by the man towards this woman.” (I_01—Gender: F, age: 38, profession: psychologist, workplace: private practice).

Narcissistic Traits

“Perhaps it is very lateral to narcissism, of the man who needs to denigrate the other to feel superior. So, he carries on this sort of “psychological” game of denigrating the other”

(I_10—Gender: M, age: 49, profession: municipality official, workplace: Campania Region office).

Low Level of Self-Esteem

“Not a monster, but a very insecure, very fragile person, poorly cultured and with a great need to be loved, but not capable of loving.” (I_24—Gender: F, age: 35, profession: psychologist, workplace: private practice).

Violence as Fear of the Other

Among the various reasons for perpetuating violence given by men, the professionals often refer to fear frequently related to different aspects: fear of difference, of loss of the partner, of her betrayal, of the loss of significant relationships, of possession over the partner, of loneliness, and partner autonomy.

“They live in an emotional dependence that brought them various fears: of losing their loved one, of being abandoned; the fear of change, of separation, detachment, loneliness creates jealousy, possessiveness, resentment, anger, guilt, self-disgust and a sense of inferiority towards the partner …” (I_29—Gender: M, Age: 70, Profession: Psychotherapist, Work Context: Courts).

### 3.2. The Representation of Gender-Based Violence Acted by Perpetrators

The categories grouped in this conceptual macro-area concerning how gender-based violence is described and experienced by interviewees, are:Disbelief Towards Domestic Violence

The interviews reveal the difficulty for professionals and victims to “digest” that violence takes place within the home.

“Home means a safe place, a safe harbor where you undress a little from all the things that defend you from by the outside world, and where you do not expect it, violence comes to you …. You are incredulous, it is not possible that it is really true and therefore there is also a resistance in recognizing it.” (I_37—Gender: F, age: 41, profession: psychotherapist, workplace: private practice).

Invisibility

Personnel emphasizes that violence is expressed through a social and relational code that makes it invisible. The perpetrator, used to this language, does not recognize its own violence.

“The woman victim reports that her partner does not understand why she reported him stating he had bad luck … The slap was almost normal for him, because his father beat his mother. And he said ‘I had bad luck, bad because she fell, her eardrum was pierced and she had to go to the hospital’ Indeed, this case shows how it is important to work on the assumption of responsibility trying to put aside the defensive attitude expressed by ‘she provokes.’” (I_32—Gender: F, age: 62, profession: psychotherapist, workplace: OLV).

### 3.3. The “Naturalness” of Gender-Based Violence

The categories grouped in this conceptual macro-area concerning how the patriarchal character of culture “justifies” violence are:Violence as a Patriarchal Male Chauvinist Expression

According to some interviewees, partner violence is inherent in its way of being, “But anything can trigger violence; there is no specific dynamic, it would be ideal if there was a specific dynamic! Therefore, there are people who express themselves through a code which is the code of violence. So, they aren’t waiting for the circumstance to arise! If there are no trigger events, they create them in their mind!” (I_22—Gender: F, age: 70, profession: psychotherapist, working context: private practice).

For professionals, a patriarchal culture based on gender asymmetry and gender stereotype [64] is at play. Most of the professionals describe an intergenerational and transgenerational transmission of violence among men and women in their reciprocal interactions.

Social Silence

The issues relating to the silence of the community, in relation to the “normality of violence”, indicate the habit and addiction to violence, widespread in every context.

“In certain contexts, violence is such a normal, accepted fact, that perhaps what appears strange is the woman who reacts. It is more surprising that the woman reacts rather than anything else.” (I_28—Gender: F, age: 37, profession: Lawyer, working context: CAV and private practice).

The Literacy of Violence

The cultural dimension was considered of central importance by the respondents, so the violence in the intimate partner relationship becomes justified and accepted.

“…Surely violence arises from a cultural fact, according to which men think they can have control over women’s lives. They think in some way that, for example, if the woman were their wife, she has not to escape their will, she has not to escape their control.” (I_14—Gender: F, age: 39, profession: psychotherapist, workplace: CAV).

Most respondents said that, behind a violent man or a woman victim of violence, there is a violent family of origin.

“Statistics tell us that these violent men have been children who have seen and experienced violence in their family.” (I_16–Gender: F, age: 42, profession: psychotherapist, work context: OLV).

Professionals, especially those with a psychoanalytic background, claimed that the perpetrators of violence have problems with their primary relationship. They talked about their separation difficulties, especially in loss management, about their tendency to attachment and their “distorted mothering model”. In a different approach, professionals rather told interviewers about their violent imprinting.

“These are always people who have experienced deprivation, deficiencies and catastrophic trauma.” (I_33—Gender: F, age: 65, profession: psychotherapist, workplace: National Health System).

Such a relation is connected with the difficulties in the separation from the primary object of love, it is related to problems of total dependency on the relative one and therefore a problem in keeping emotional independence will always be there for them [65].

Some professionals think the opposite:

“There is no common denominator. I have seen family violence resulting from the family of origin and perpetrated in the current family and I have seen violent families that have generated children who have rejected violence and said ‘I don’t want to have anything in common with this story’... it was a reaction in the opposite direction.” (I_50—Gender: F, age: 33, profession: social worker, workplace: center for families).

The Normality of “Violent Men”

According to the initial data that emerged from the interviews, the representation of the perpetrators of gender-based violence is non-stereotyped. Therefore, as for women, there would be no specific social classes involved, nor generalizations.

“Violent behavior is transversal; for example, we have seen men who did not have great culture and men who had great culture; indeed, there is not originally a common trait in violent behavior or something that unites violent men. They can be anyone, he can be the good father of a family that transforms himself when he feels he no longer has possession of his wife, or he may be the one who may be, from the beginning has been violent.” (I_14—Gender: F, age: 39, profession: psychotherapist, workplace: CAV).

### 3.4. The Possibility of Change for Gender-Based Violence Perpetrators

The categories grouped in this conceptual macro-area concerning the possibility of the perpetrator to change, are:The Perpetrator’s Impossibility to Change

However, concerning the assessment of a possible change in the men, the interviewees declare “distrust” and “resignation”.

“So, if this miracle happened…” (I_21—Gender: F, age: 60, profession: sociologist, work context: CAV).

“I believe that violent behavior is a kind of compulsion to repeat, that even men aren’t able to manage it. I am not sure that men completely intentionally decide to act that type of behavior” (I_40—Gender F. Age 27; profession: social worker, workplace: CAV).

This is a very strong issue emerging from the interviews of the professionals. It expresses their impotence in dealing with the phenomenon.

Fatherhood as an authentic motivation for change

Although resigned, professionals recognize in men some potential motivation for treatments mostly related to fatherhood:

Fatherhood is defined as a “double edged sword”, because it can also become a motivation for violence (“the beginning of violence often coincides with the beginning of a pregnancy”, as many said), an object of emotional threat, an instrument of control over women becoming an instrument of control especially when there are children under 18; most of the interviewees argued that men who act violently towards their partner automatically cannot be a good father. On the other hand, the fear of “losing” the child can lead a man to change, to improve.

“Fatherhood is a gap in some way...where you can insert a bit of change … at least you can try” (I_27—Gender: F, age: 58, profession: social worker, workplace: Local health service).

External and Internal Motivation to Change

Change is related to different motivations:

External motivations: including the fear of possible penalties (going to prison), loss of parental authority, fear of losing the partner.

“In the meantime, in my opinion it would be necessary and fundamental for them to admit, so they were aware of what happened, so they put themselves on a level of responsibility, in order to recognize the perpetrated violence. Unfortunately, they often follow treatment only because it is recommended/imposed by the judicial authorities” (I_14—Gender: F, age: 39, profession: psychologist, workplace: CAV).

Internal motivations: awareness, recognition of one’s malaise (also thanks to others) and work towards the achievement of well-being.

“I think that the strongest motivation is the one you have in wanting to change” a Psychologist (I_24—35 years old, CAV) affirmed.

Surely, spontaneous requests for help from a man can give hope for a better change.

The Table 2 illustrates all the codes selected in the study, with their respective categories and macro-categories.

## 4. Discussion

From the collected data, what emerges is that the “normality” of violence in patriarchal culture leads a man to perform violent acts and, above all, not to be able to recognize his own emotions and those of others. Professionals often told interviewers about silence within families and the whole of society.

The woman’s rejection of social gender roles is considered another aspect that could explain the man’s violence against his partner, especially regarding submission and subordination.

The professionals reported that perpetrators of gender-based violence talk about a lack of care from their partners and for this reason they qualify women as “bad wives and bad mothers”.

The literature also confirms these data [26,37] and one of the main obstacles between the perpetrator of violence and treatment frequently lies in the attitude of blaming the victim and denigrating the relationship with her. Indeed, the man releases his frustration through aggressive attitudes and control.

Other risk factors are marital separation or often pregnancy. The latter could represent the starting or increasing moment of domestic violence, as confirmed by literature [5,66,67,68]. A possible explanation for this common phenomenon would concern the concurrence of the man’s need to receive exclusive love from the woman and his excessive jealousy towards the child she carries in her womb and the weakness and fragility of the pregnant woman.

Since childhood, males are denied some emotions and therefore the only choice they have is to use violence and anger, even in situations of conflicting relationships [69].

For the professional, violent men in intimate relationships lack “emotional literacy”: they are not able to give a name to emotions, even negative ones, because they have not been used to doing it. When they feel the armor of masculinity wavering—the armor that makes them strong and invincible—they attack their partner because, through violent action, they have the feeling of having restored order and become “real men” again. Then, the use of violence becomes a wrong choice, but still a choice, to solve and silence relationship problems [70].

Our professionals do not have a representation of man as a “monster” and a true “typical” perpetrator has not been identified. According to both the interviews and literature, all men are potentially perpetrators of violence. The perpetrator “is the neighbor”, he is unsuspected. However, common characteristics of violent men have been reported: traits of pathological fragility and narcissism, seductiveness and manipulation (also towards the professionals), communication and thought difficulties, history of witnessed or direct violence during childhood, low tolerance to frustration, obsessive compulsive and sociopathic traits.

Concerning the possible change in men, most of the interviewed professionals were “resigned” and skeptical.

Nevertheless, possible motivations for taking a treatment path have been identified: fatherhood, fear of losing one’s partner, awareness or fear of possible penalties.

Both the interviews and literature confirm that children can be a motivation for a man to change, but at the same time, they can be a tool in the hands of men to continue perpetrating violence, as a recourse against one’s partner [40,71,72,73,74,75].

According to the interviewees, during the treatment with a perpetrator, it is considered to be very important to focus on empathy, making the authors put themselves in the shoes of women and children.

Other aspects that are emphasized in the treatment of perpetrators are as follows: always having a non-judgmental attitude and being able to embrace their fears and work on the ability to recognize and think of one’s own and/or others’ emotions in order to improve their communication skills and to make them aware of their internal stereotypes (working on their personal history).

Furthermore, as Ferrer-Perez et al. [76] suggest, a flexible attitude of professionals to adapt the treatment program to the characteristics of the perpetrators is of significant importance, assuming that the relationship with professional and perpetrator is very important to promote a change [77].

In conclusion, the explanatory Core Category resulting from the analysis of the interviewees is “Inescapable repetitiveness of violent behavior” (see Figure 2). Why? Because what emerges from the interviews is that while the woman can get out of her “role” as a victim, for the man, instead, the change is considered impossible: only a miracle can make it possible.

According to the participants, the woman has the possibility of freeing oneself from subordination, while, the man is condemned to his inevitable and immobile repetitiveness of violence.

The Figure 2 illustrates the core category and the main related concepts.

## 5. Conclusions

The research aimed to focus on the perception of perpetrators as a useful element in designing innovative intervention policies. In fact, as long as public policies only concern women, we cannot even think that the problem has been solved, but will only be shifted, passed on to the next victim. We have seen how gender-based violence has historical, cultural and social roots and the contribution of professionals to change this can be of the utmost importance.

From listening to the interviews, unresolved issues can be deduced. A silence emerges, as a turning of one’s gaze to the other side, because, after all, “the perpetrator of violence is comparable to the monster whose fate there is no interest in; he just has to pay”. But this is not enough.

The core category focuses on an inevitable repetitiveness. Even if the professionals seem in favor of “recovery” interventions for the perpetrators of violence, there seems to be a “paralyzing essence” that leads them to always stay one step behind regarding this theme, either caused by prejudice or by disinterestedness.

These results detect attitudes of social health personnel and their feelings of impotence towards gender-based perpetrators; this reflective knowledge can help to develop the best transformative attitudes toward the phenomenon.

Conversely, the knowledge of psychological and psychopathological correlates of male violence, highlighting their “naturality”, could lead to the improvement of the current services’ strategies aimed at supporting men’s motivation for treatment.

The knowledge of the predictive factors of male violence, as well as of the risk and protective factors and of the reasons behind the violent behavior, will help define new intervention strategies.

Moreover, both the interviews and literature [78] show a severe level of mental distress experienced by these men in taking charge of their psychophysical health and an inability to express themselves, but these experiences of suffering mirror the feelings of impotence experienced by the professionals.

In conclusion, despite the fact that the study found that there is no “typical” violent man and despite the increased awareness of the heterogeneity of gender-based perpetrators [79], these professional ‘vignettes’ help us in detecting new strategic interventions.

One important aim is the taking care of the professionals and their training, while one of the intervention objectives is to find a part inside the perpetrator with which to be able to ally, in order to go beyond violence, beyond the categorization of the violent man and be able to get in touch with the suffering of the person.

The feelings of fear and anger of the staff towards the perpetrators as well as their collusion with the “denial modality” of the perpetrators of violence could lead professionals to develop an attitude of refusal towards them, on the contrary, to minimize their violent acts as previously significantly highlighted by Creazzo [46] and Esposito et al. [80]. The criminalization of perpetrators and a social unawareness about gender-based violence are in fact the two ineffective options assumed by the literature [46] behind this helpless repetitiveness. The personnel’s reflectivity, the pursuit of knowledge, empathy and new cultural visions for men–women relations are the pillar of social change.

In fact, it is urgent to establish an active and convinced alliance with the healthy part of the man. As pointed out by Boira et al. [77], it is necessary to increase perpetrators’ trust in services to induce them to request support and treatment.

A vision based on the mistrust of any possible change makes it impossible to get out of a social policy where the solution is once again required only of women.

The limit of this research is that it was realized in a situated context; therefore, its results are not to be generalized to different contexts; the participants are mostly women, as, in fact, is the composition of personnel working in such services, but it would be interesting to analyze male professionals’ attitudes more closely and be able to define different attitudes more precisely. Moreover, the work experience of our personnel range from 1 to 45 years and this element could be a significant variable concerning their training and individual professional career path. Lastly, we have to consider that our aim was to collect experience of different professionals dealing with gender-based violence. However, the richness of our variables while offering a large frame of analysis, have not allowed us to define the effects of specific trainings and backgrounds. From the quotations, we can observe that the psychotherapeutic approach is the less judgmental one; however, there is not a discussion about the implication in service organizations and treatments related to this approach. For further research, it would be interesting to propose to organize some focus group aimed at gathering specific attitudes related to the different training approaches, namely, psychoanalytic, systemic and community psychology. The research was aimed at examining attitudes and representations of professionals of services dealing in fighting violence, but a more direct analysis of the motivations, representations and actions of the perpetrators of violence would add a strong complementary vision.

In order to avoid investigating the phenomenon of violence in its extreme dimensions, we have not interviewed sex offenders detained for femicide and injuries against the partner, but an objective for future studies are direct interviews with men at the first report of violence against their partners. The aim of further research will be to interrogate the perpetrators of violence who turn to specific centers, leading to the definition of strategies for possible cultural and organizational intervention, including men in fighting the phenomenon.

## Figures and Tables

**Figure 1 ijerph-17-07061-f001:**
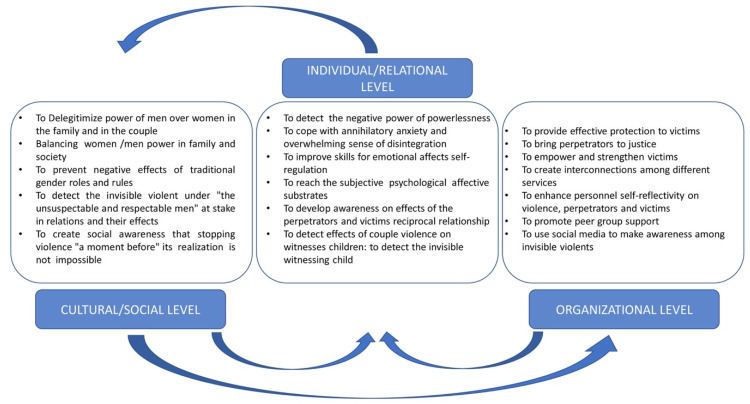
The ecological model and operationalization of goals and actions at different levels. Source: Di Napoli et al., 2019 [10].

**Figure 2 ijerph-17-07061-f002:**
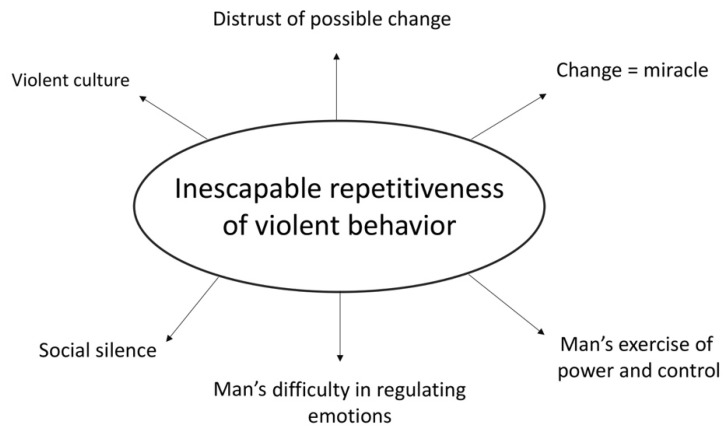
Core category and semeiotic network “Inescapable repetitiveness of violent behavior”.

**Table 1 ijerph-17-07061-t001:** Description of participants.

Codes Interview	Age	Gender	Profession	Work Context	Years of Work	Years of Work with Violence Cases	Work with Perpetrator
I_01	38	F	Psychologist	Private Practice	From 6 to 10	From 6 to 10	Yes
I_02	34	F	Psychologist	Antiviolence Center CAV	From 6 to 11	From 1 to 5	No
I_03	37	F	Psychologist and Psychotherapist	Antiviolence Center CAV	From 6 to 12	From 1 to 5	No
I_04	46	F	Assistant councilor for equal opportunities	Office of Campania Region	From 1 to 15	From 1 to 5	No
I_05	61	F	Psychologist and Psychotherapist	Center for Violent Men OLV	Over 15	Over 15	Yes
I_06	55	F	Emergency Doctor (Campania Region)	Hospital	Over 15	Over 15	No
I_07	53	F	Social Worker	Center for Families	Over 15	From 6 to 10	Yes
I_08	31	M	Psychologist and Psychotherapist	Private Practice	From 1 to 5	From 1 to 5	No
I_09	66	F	Psychologist and Psychotherapist	Center for Violent Men OLV	Over 15	From 6 to 10	Yes
I_10	49	M	Official in Department of Equal Opportunities	Office of Campania Region	From 11 to 15	From 6 to 11	No
I_11	52	F	Nurse	Center for Violent Men OLV	Over 15	From 1 to 5	No
I_12	41	F	Psychologist and Psychotherapist	Antiviolence Center CAV	From 6 to 10	From 6 to 10	No
I_13	46	F	Equal opportunities office manager	Department of Equal Opportunities	From 11 to 15	From 11 to 15	Yes
I_14	39	F	Psychologist and Psychotherapist	Antiviolence center CAV	From 11 to 15	From 6 to 10	No
I_15	44	F	Social worker	Services for Minors and Families	Over 15	Over 15	Yes
I_16	42	F	Psychologist and Psychotherapist	Antiviolence Center CAV	Over 15	From 1 to 5	No
I_17	43	F	Lawyer	Antiviolence Center CAV	Over 15	From 1 to 6	No
I_18	30	F	Psychologist	AntiviolenceCenter CAV	From 6 to 10	From 6 to 10	No
I_19	42	F	Lawyer	Forensic Association and Antiviolence Center	From 11 to 15	From 6 to 10	Yes
I_20	43	F	Lawyer	Antiviolence center CAV	From 11 to 15	From 1 to 5	No
I_21	60	F	Sociologist	Antiviolence center CAV	Over 15	Over 15	No
I_22	70	F	Psychologist and Psychotherapist	Private Practice	Over 15	Over 15	Yes
I_23	59	F	Psychologist	Center for Families	Over 15	Over 15	Yes
I_24	35	F	Psychologist and Psychotherapist	Antiviolence Center CAV	From 6 to 10	From 6 to 10	No
I_25	31	F	Psychologist and Psychotherapist	Center for violent men OLV	From 6 to 10	From 6 to 11	Yes
I_26	32	F	Psychologist and Psychotherapist	Center for Violent men OLV	From 1 to 5	From 1 to 5	No
I_27	58	F	Social worker	Health Consultancy	Over 15	Over 15	Yes
I_28	37	F	Lawyer	Antiviolence Center CAV	From 6 to 10	From 6 to 10	No
I_29	70	M	Psychologist and Psychotherapist	Court of Italy	Over 15	Over 15	Yes
I_30	62	F	Social worker	Center for Families	Over 15	Over 15	Yes
I_31	57	F	Psychologist	Center for Families	Over 15	Over 15	Yes
I_32	62	F	Psychologist and Psychotherapist	Center for Violent Men OLV	Over 15	Over 15	Yes
I_33	65	F	Psychologist and Psychotherapist	Local Health Service (ASL)	Over 15	Over 15	Yes
I_34	41	F	Councilor for Equal Opportunities	Office of Campania Region	Over 15	From 1 to 5	No
I_35	29	F	Psychologist	Center for Violent Men OLV	From 1 to 5	From 1 to 6	Yes
I_36	55	F	Psychologist	Center for Violent Men OLV	Over 15	From 6 to 10	Yes
I_37	41	F	Psychologist and Psychotherapist	Private Practice	From 6 to 10	From 1 to 5	Yes
I_38	46	F	Lawyer	Court of Naples	From 6 to 11	From 6 to 10	No
I_39	56	F	Psychologist and Psychotherapist	Center for Violent Men OLV	Over 15	From 6 to 10	Yes
I_40	27	F	Social worker	Antiviolence Center CAV	From 1 to 5	From 1 to 5	Yes
I_41	43	F	Lawyer	Court of Naples	Over 15	Over 15	Yes
I_42	28	F	Psychologist	Antiviolence Center CAV	From 1 to 5	From 1 to 5	No
I_43	38	F	Psychologist	Social Promotion Association	From 6 to 10	From 6 to 10	No
I_44	31	F	Psychologist	Antiviolence Center CAV	From 1 to 5	From 1 to 5	No
I_45	35	F	Psychologist	6 Private Practice	From 1 to 5	From 1 to 5	No
I_46	31	F	Psychologist and Psychotherapist	Center for Violent Men OLV	From 6 to 10	From 6 to 10	No
I_47	61	F	Psychologist and Psychotherapist	Health Consultancy	Over 15	From 6 to 10	No
I_48	35	F	Psychologist and Psychotherapist	Antiviolence Center CAV	From 6 to 10	From 1 to 5	Yes
I_49	58	M	Psychologist	Center for Families	Over 15	Over 15	Yes
I_50	33	F	Social Worker	Center for Families	From 1 to 5	From 1 to 5	No

**Table 2 ijerph-17-07061-t002:** Table illustrates all the codes selected in the study, with their respective categories and macro-categories.

Codes	Categories	Macro-Categories
1. Difficulty Thinking about Emotions	- Difficulty Expressing Themselves and Their Emotions	The Representation of the Perpetrator
2. Man Cannot Manage Emotions	
3. Man’s Impulsiveness	
4. Man Cannot Manage Conflictual	
Relationships	
5. Man who Betrays	
6. Man’s Vulnerability	
7. Drug Addiction	
8. Communication Problem	
9. Repetitiveness of Man’s Violent Behavior	
10. Lack of Empathy as a Defense Mechanism	
11. Methods of Releasing Tensions	
12.Violence as a Way of Releasing Tensions	
13. Man’s Obsessive Personality	
14. Perversion of Man	- Compulsive Obsessive Traits
15. Man’s Paranoia	
16. Sociopathy	
17. Man’s Compulsion to Repeat	
18. Compulsion of Man	
9. Repetitiveness of Man’s Violent Behavior	
19. Woman as a Manipulable Object	
20. Anguish of Abandonment of Man	
21. Man’s Dependence	- Manipulative Behavior
22. Constant Search for Confirmation (Man)	
23. Jealousy of Man	
24. Manipulative Man	
25. Abuse of Man	
26. Exercise of Power	
27. Men who Show Themselves as Blue Princes	
28. Denigration of Women	
29. Isolation of the Woman	
30. Possession of the Woman by the Man	
31. Woman Deprived of Liberty	
32. Feeling of Possession	
33. Overwhelm	
34. Man Unable to Manage	
Women’s Emancipation	
35. Very Controlled Behavior	
36. Control Through Taxation	
37. Economic Control	
38. Control by Man	
39. Stalking	
40. Violence as a Woman Correction	
41. Manipulative Man	
42. Emotional Violence	
43. Verbal Violence	
44. Denigration of Women in the Presence of their Children	
45. Code of Violence	
46. Man’s Identity Structured on Woman’s Identity	
47. Feeling of Possession	
48. Very Controlling Behavior	
49. Man’s Narcissism	
50. Seductive Man	- Narcissistic traits
51. Man Reluctant to Treatment	
52. Man’s Lack of Empathy	
53. Destructive Dynamics	
54. Projective Identification	
20. Anguish of Abandonment of Man	
21. man’s dependence	
22. constant search for confirmations (man)	
55. discomfort	
56. Low self-esteem	- Low level of self-esteem
57. Impotence	
58. insecurity	
59. dissatisfaction	
60. weakness	
61. feeling of emptiness	
62. loneliness	
63. fragile man	
64. man victim of himself	
65. Shame of man	
6. man’s vulnerability	
66. separation difficulties	
67. fusionality	
68. Projective identification	
46. man’s identity structured on woman’s identity	
20. anguish of abandonment of man	
21. man addiction	- Violence as fear of the other
22. constant search for confirmation (man)	
16. sociopathy	
66. separation difficulties	
67. fusionality	
69. danger of loss	
70. violence as a family mandate		The Representation of Gender-Based Violence Acted by Perpetrators
71. non-admission of violence	
72. justification of violence by man	
73. unawareness of man	
74. it is difficult to accept that violence takes place within the home	- Disbelief towards domestic violence
75. social collusion on the man-woman relationship	
76. isomorphism between society and violent relationship	
77. invisibility of the phenomenon	
78. patriarchy	- Invisibility
79. woman property of man for culture	
80. on a social level, the diagnosis removes responsibility for man	
75. social collusion on the man-woman relationship	
78. patriarchy	- Violence as a patriarchal male chauvinist expression	The “Naturalness” of Gender-Based Violence
79. woman property of man for culture	
75. social collusion on the man-woman relationship	
70. violence as a family mandate	
45. Code of violence	
81. responsibility of society	
82. family of origin that does not support the woman	- Social silence
83. silence of the family	
75. social collusion on the man-woman relationship	
84. woman’s education	
85. man’s education	- The literacy of violence
86. gender stereotypes	
87. intergenerational transmission	
70. violence as a family mandate	
82. family of origin that does not support the woman	
88. insane family of origin	
89. families of origin that do not teach empathy	
90. lack of release from family of origin	
83. silence of the family	
79. woman property of man for culture	
75. social collusion on the man-woman relationship	
91. cultural justification of violence	
92. “normality” of violence in the patriarchal culture	
83. silence of the family	- The normality of “violent men”
93. silence of the society	
78. patriarchy	
86. gender stereotypes	
79. woman property of man for culture	
91. cultural justification of violence	
45. code of violence	
94. man’s lack of empathy towards children	- The perpetrator’s impossibility to change	The Possibility of Change for Gender-Based Violence Perpetrators
95. man unaware of the harm inflicted on his children	
79. woman property of man for culture	
80. on a social level, the diagnosis removes responsibility for man	
85. man’s education	
91. cultural justification of violence	
96. men aware of aggression but blame the other	
97. fatherhood as a motivation for change	- Fatherhood as an authentic motivation for change
98. fatherhood as a motivation for change	- External and internal motivation to change
99. fear of man	
100. loss of parental authority	
101. destruction of ties	
102. awareness of man	
103. intrinsic motivation	
104. achieve personal well-being	
105. recognition of pain by man	
106. feeling not judged can help man to change	
107. avoid punishment (motivation for man to change)

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
