# Peer review of "Downside: The Perpetrator of Violence in the Representations of Social and Health Professionals"

_ijerph, 2020, doi:10.3390/ijerph17197061_

Round 1
Reviewer 1 Report
Overall, nice study and in the current COVID-19 environment, where families and couples are shut in together, the risk of family violence increases. Thus, the relevance of this study. I have a few suggestions:
Page 1
Lines 37-38 -- I would combine this sentence with next paragraph
line 44 -- where is IPV defined?
Page 2 -- the two paragraphs (lines 55-59) should be combined and better integrated to give an introduction to the topic of the roots of violence inflicted by men
Page 5 -- Last few paragraphs (lines 183-194) need to be better integrated into a concluding paragraph for the background section. It is confusing.
The overall goal of the study is a little unclear. Also, should put 'study aim' in the last paragraph of background section and not in the methods section.
Page 6 -- a little more information on Grounded Theory Methodology and how that method was used would be helpful. Spent so much time on background/lit review and not much on methodology.
In Results section, authors mention that codes were grouped into several categories and macro-categories; then author lists out codes and quotes. It would be helpful for reader if codes, macro-codes, and then sample quotes were presented in a table format.
Page 10 (Discussion) -- I am a little confused. In first sentence of discussion (line 434), authors state "....the normality of violence in patriarchal culture leads a mans .....". This is assuming that the culture you studied is in fact patriarchal. I might have missed it, but I don't remember anywhere in the introduction/lit review where authors stated/established that the culture/participants being studied (Naples, Italy) is in fact 'patriarchal'. This was not established clearly. I bring this up because other international locations/cultures and not patriarchal in nature but in fact are matriarchal -- such as many Pacific cultures.
Author Response
- Lines 37-38 -- I would combine this sentence with next paragraph.
THANK YOU FOR YOUR SUGGESTION, WE HAVE USED THIS SENTENCE TO INTRODUCE THE NEXT PARAGRAPH IN A CONSISTENT WAY. WE KEPT IT.
- Line 44 -- where is IPV defined?
WE EXPLAINED THE ACRONYM OF IPV (SEE NOW LINE 46)
- Page 2 -- the two paragraphs (lines 55-59) should be combined and better integrated to give an introduction to the topic of the roots of violence inflicted by men
THANK YOU, WE COMBINED THE TWO PARAGRAPHS (SEE NOW LINES 57-61)
- Page 5 -- Last few paragraphs (lines 183-194) need to be better integrated into a concluding paragraph for the background section. It is confusing.
WE REFORMULATED THE LAST FEW PARAGRAPHS, AS YOU SUGGESTED (SEE NOW LINES 196-209).
- The overall goal of the study is a little unclear. Also, should put 'study aim' in the last paragraph of background section and not in the methods section.
WE DID IT, AND WE ADD ALSO THE FOLLOWING: “THERE IS THE NEED TO DEEPEN HOW THEIR REPRESENTATIONS INFLUENCE THE VISION OF THE PHENOMENON AND THEIR ATTITUDES AND ALSO TO UNDERSTAND IF THEIR VISIONS OF THE PERPETRATORS AFFECT THE SERVICES’ INTERVENTION STRATEGIES AND ENHANCE THE CULTURAL COMPREHENSION OF THE PHENOMENON” (SEE NOW LINES196-209).
- Page 6 -- a little more information on Grounded Theory Methodology and how that method was used would be helpful. Spent so much time on background/lit review and not much on methodology.
WE HAVE INTEGRATED THE PARAGRAPH “PROCEDURES AND METHODS” (SEE NOW LINES 264-265, 270-280, 285-286).
In Results section, authors mention that codes were grouped into several categories and macro-categories; then author lists out codes and quotes. It would be helpful for reader if codes, macro-codes, and then sample quotes were presented in a table format.
WE ADDED NOW IN THE TEXT THE TABLE 2 (pp. 11-15).
- Page 10 (Discussion) -- I am a little confused. In first sentence of discussion (line 434), authors state "...the normality of violence in patriarchal culture leads a mans ...". This is assuming that the culture you studied is in fact patriarchal. I might have missed it, but I don't remember anywhere in the introduction/lit review where authors stated/established that the culture/participants being studied (Naples, Italy) is in fact 'patriarchal'. This was not established clearly. I bring this up because other international locations/cultures and not patriarchal in nature but in fact are matriarchal -- such as many Pacific cultures.
WE SUPPOSED THAT OUR REFERRING TO BORDIEU THOUGHTS WOULD BE ENOUGH, BUT NOW WE INTRODUCED THE FOLLOWING IN BACKGROUND: “THE RESEARCH IS SITUATED IN SOUTH ITALY, A CONTEXT WHERE A PATRIARCHAL CULTURE IS STILL VERY DEEPLY ROOTED IN MEN WOMEN RELATIONSHIPS (LINES 200-201).
Reviewer 2 Report
In relation to the different sections of the paper, there are some aspects that it would be important to modify in order to improve it.
Introduction:
It is interesting to include some more recient research about batterer treatment programs in Europe. For instance:
https://journals.sagepub.com/doi/abs/10.1177/0306624X16673853
https://www.work-with-perpetrators.eu/research/project-wwp/results
And, above all, some specific research related to the opinion of professionals who work in this type of program and who offer interesting information to complete the discussion. For instance:
https://www.sciencedirect.com/science/article/pii/S1132055913700152
It is suggested to carefully review the statements on Johnson's model (lines 96-107) since it is a proposal that can be very useful for understanding intimate partner violence against women from a feminist perspective, and that, if it is questioned, it is precisely because of its feminist character (not because it does not take into account cultural aspects).
Method and Results
In Method section it is essential to include a table with the characteristics of the participants and with a code for each one of them.
If the list of 50 people is considered too long, the list with codes can be included as supplementary information, but it is essential.
This code should be used later in the results section to identify each verbatim. This will achieve two objectives: to make the results section easier to read, and to check that the statements have been made by different participants (and not all of them by the same participants).
Discussion and conclusions
It is essential to include some limitations of the research that have not been made explicit in the paper:
1) All qualitative research is limited by nature (not generalizable).
2) The gender composition of the sample (45 women and 5 men) can introduce some distortions in the results (the attitudes of men and women towards violence against women are often different).
3) The huge difference in professional experience between the professionals interviewed (from 1 year of experience to 45 years of experience) can also be a source of bias (people who are older may be more disappointed, may have different training, etc.).
4) The professional profiles of the people interviewed are very diverse.
The abstract states: "Open interviews were carried out with welfare and 17 health professionals and the Grounded Theory Methodology was used to analyze the collected data". These diversity can significantly condition the results obtained (because their training is very diverse, because they do not all have the same level of responsibility in intervention programs, etc.).
Therefore, all these limitations should be mentioned in the article, and their implications discussed in the corresponding section.
Author Response
Comments and Suggestions for Authors
In relation to the different sections of the paper, there are some aspects that it would be important to modify in order to improve it.
Introduction:
- It is interesting to include some more recent research about batterer treatment programs in Europe. For instance:
https://journals.sagepub.com/doi/abs/10.1177/0306624X16673853 -
https://www.work-with-perpetrators.eu/research/project-wwp/results
And, above all, some specific research related to the opinion of professionals who work in this type of program and who offer interesting information to complete the discussion. For instance:
https://www.sciencedirect.com/science/article/pii/S1132055913700152
THANKS, WE ADDED THE REFERENCES SUGGESTED IN THE SECTION INTRODUCTION (LINES 36- 38) AND IN THE SECTION DISCUSSION (LINES 506-508). PLEASE SEE NOW IN YELLOW
- It is suggested to carefully review the statements on Johnson's model (lines 96-107) since it is a proposal that can be very useful for understanding intimate partner violence against women from a feminist perspective, and that, if it is questioned, it is precisely because of its feminist character (not because it does not take into account cultural aspects).
SEE NOW THE FOLLOWING: “IN OUR VISION, HOWEVER, ALSO CONFLICTS THAT ARISE IN THE COUPLE BETWEEN PARTNERS ARE FREQUENTLY INSCRIBED WITHIN A PATRIARCHAL FRAME EVEN WHEN THEY RARELY LEAD TO SERIOUS FORMS OF VIOLENCE ,ALTHOUGH THIS APPROACH INDUCES US TO TAKE INTO ACCOUNT THE INTERACTION AND ESCALATION OF FORMS OF VIOLENCE MORE THAN ITS INCIDENCE AND PREVALENCE IT IS A WAY TO DETECT THE MORE SERIOUS CASES AND THEIR DANGER TO WOMEN’S LIVES, AND AT THE SAME TIME IT MAY BE THE BACKGROUND TO CREATE INTERVENTION STRATEGIES DIRECTED TO SEVERE VIOLENCE PERPETRATORS DETECTING PRECOCIOUSLY THE DANGERS OF THEIR BAHAVIOUR FOR WOMEN AND CHILDREN (SEE NOW LINES 102-108).
IN YELLOW WHAT WE ADD, IN RED WHAT WE ERASED.
THE SENTENCED ERASED IS “BUT IN PRINCIPLE IT DENIES THE CULTURAL DIMENSIONS AT PLAY IN VIOLENCE BY INDUCING A DIFFERENCE BETWEEN VIOLENCE AND CONFLICTS THAT MAY CAUSE SERIOUS PROBLEMS RATHER THAN PROVIDING A SOLUTION TO UNDERSTANDING THE PHENOMENON”
Method and Results
- In Method section it is essential to include a table with the characteristics of the participants and with a code for each one of them.
If the list of 50 people is considered too long, the list with codes can be included as supplementary information, but it is essential.
WE ADDED THE TABLE 1 (p. 6).
- This code should be used later in the results section to identify each verbatim. This will achieve two objectives: to make the results section easier to read, and to check that the statements have been made by different participants (and not all of them by the same participants).
WE DID IT NOW, SEE THE QUOTATIONS IN THE WHOLE TEXT.
- Discussion and conclusions
It is essential to include some limitations of the research that have not been made explicit in the paper:
1) All qualitative research is limited by nature (not generalizable).
2) The gender composition of the sample (45 women and 5 men) can introduce some distortions in the results (the attitudes of men and women towards violence against women are often different).
3) The huge difference in professional experience between the professionals interviewed (from 1 year of experience to 45 years of experience) can also be a source of bias (people who are older may be more disappointed, may have different training, etc.).
4) The professional profiles of the people interviewed are very diverse.
The abstract states: "Open interviews were carried out with welfare and 17 health professionals and the Grounded Theory Methodology was used to analyze the collected data". This diversity can significantly condition the results obtained (because their training is very diverse, because they do not all have the same level of responsibility in intervention programs, etc.).
Therefore, all these limitations should be mentioned in the article, and their implications discussed in the corresponding section.
WE CONSIDERED YOUR SUGGESTION AS YOU CAN SEE IN THE TEXT IN YELLOW (LINES 562-577).